# Research on Knowledge Management Models at Universities Using Fuzzy Analytic Hierarchy Process (FAHP)

**Ngoc Thach Pham [1], Anh Duc Do [2],\*, Quang Vinh Nguyen [3], Van Loi Ta [2], Thi Thanh Binh Dao [4], Dieu Linh Ha [5] and Xuan Truong Hoang [6]**

1   Board of Trustees, Hanoi University, Hanoi 12011, Vietnam; thachpn@hanu.edu.vn
2   School of Trade and International Economics, National Economics University, Hanoi 11616, Vietnam; loitv@neu.edu.vn
3   Faculty of Business Administration, University of Labour and Social Affairs, Hanoi 11313, Vietnam; quangvinh191081@gmail.com
4   Faculty of Tourism and Management, Hanoi University, Hanoi 12011, Vietnam; binhdtt@hanu.edu.vn
5   Ministry of Education and Training, Hanoi 11615, Vietnam; dieulinhha83@gmail.com
6   Department of Finance and Accounting, Hanoi University, Hanoi 12011, Vietnam; truonghx@neu.edu.vn
\*   Correspondence: ducda@neu.edu.vn

**Abstract:** This study aims to investigate and evaluate factors related to the knowledge management model at universities in Hanoi, Vietnam. Based on the system literature review (SLR) approach, the study follows descriptive and inductive approach results of the document review process. Eight factors were synthesized with the fuzzy analytic hierarchy process (FAHP) to evaluate the priority order. Ten experts from seven universities participated in the survey. The results rank as follows: (1) knowledge sharing factor (this also has the highest best nonfuzzy performance (*BNP*) and average multiplier weight (GM)); (2) knowledge management with big data systems; (3) knowledge creation; (4) knowledge use; (5) knowledge gathering; (6) leadership; (7) knowledge rating; and (8) knowledge storage. Discussions, conclusions, limitations of the study, and suggestions for future studies are also mentioned in this study.

**Keywords:** knowledge management; Hanoi; system literature review; fuzzy analytic hierarchy process



## 1. Introduction

Knowledge is an important force in driving performance and creating a competitive advantage in an organization. According to Evers and Gerke [1], since its introduction in the 1980s, knowledge management has become a standard practice to measure organizations on a global scale. In Vietnam, knowledge management and knowledge economy are often mentioned in the documents of the Party and State, specifically in the socioeconomic development plan of ministries, branches, and localities. However, the level of understanding surrounding knowledge management in Vietnam, including both theoretical and practical levels, remains limited.

In addition to teaching, university lecturers have the important task of scientific research. Knowledge management plays an important role in the university, both as a basic function and contributing factor to the school's branding. In recent years, research on knowledge management has been carried out around the world [2–4]. Research has focused on problems related to acquiring, creating, storing, sharing, developing, diffusing, and deploying knowledge [5–7] by individuals or teams/groups within an organization to improve efficiency. Meanwhile, new topics related to knowledge management systems are being discussed, including community of practice (CoP) and big data.

The main objective of this study is to collect, build, and synthesize a complete, systematic document review of current issues related to the field of information management and information management models in universities. In addition, the study aims to verify

the main factors in the knowledge management model used in universities in Vietnam. The study used a system literature review (SLR) approach and followed a descriptive and inductive approach to identify and discuss major issues in the field of knowledge management in universities.

Next, the study used the fuzzy analytic hierarchy process (FAHP) method [8] to solve complex decision-making problems with different selection criteria and people involved in the decision-making process. Although the conventional AHP explains and describes expert knowledge, it cannot detail or reflect human behavior and thinking [8]. Therefore, FAHP, a fuzzy extension of AHP, was developed to solve fuzzy gradation problems. The matrix pair comparisons in the FAHP process are fuzzy numbers, allowing the decision maker the ability to assign priorities in the form of a natural linguistic expression of the importance of each criterion [9]. Consequently, fuzzy logic provides a systematic basis for dealing with ambiguous or undefined situations [10]. Also, the applications of fuzzy decision making in various fields such as Ali et al., (2019) in software defined networking for controller selection and controller placement, Lyu et al., (2020) in risk assessment using a new consulting process or determining the importance of the criteria of traffic accessibility (Stanković, et al., 2019) [11–13]. Related to knowledge management subject, fuzzy decision-making methods is also applied in: Albooyeh & Yaghmaie (2019) in evaluation of knowledge management model in construction companies; Sani et al., (2019) in Knowledge management adoption to financial institutions and Yazici et al., (2020) in determine the tacit knowledge criteria [14–16]. Sani et al., (2019) state that although fuzzy AHP requires tedious computations, it can capture a human's appraisal of ambiguity when complex multi-attribute decision-making problems are considered in knowledge management [15].

The following research questions have been identified:

1.  What factors relate to the knowledge management model in universities?
2.  What is the ranking of these factors in the knowledge management model?

Prioritized factors help managers in higher education institutions recognize key aspects of success in knowledge management. This study aims to contribute to the research by exploring the priorities of these factors in the knowledge management context of universities in Vietnam.

## 2. Theoretical Basis

### 2.1. Knowledge and Knowledge Management

Among the definitions of knowledge management, Smith [17] has a high degree of coverage and is used in many studies. According to Smith [17], knowledge management is: (1) a process of collecting and accumulating knowledge for an organization; (2) organizing, distributing, and applying knowledge to an organization's activities; (3) sharing and protecting the interests of knowledge creators; and (4) taking measures to motivate employees to maintain valuable knowledge for an organization. The main activities in knowledge management include:

- Searching, accumulating, and sharing knowledge in an organization
- Motivating employees
- Converting and distributing knowledge among the team
- Protecting the rights of knowledge creators

Miltiadis et al. [18] showed that researchers divide knowledge management into four components and steps: (1) collecting; (2) transferring; (3) applying; and (4) protecting and/or preserving knowledge. However, Mehta [19] argued that it is necessary to place knowledge management in other activities of an organization. Similarly, Lee et al. [20] described knowledge management in parallel with factors such as culture, information technology, strategy, leadership style, and motivational tools in the creation of knowledge products. Bhatti et al. [21] and Rašula et al. [22] showed that increasing the implementation of knowledge management activities will increase the organizational efficiency of

enterprises. According to Du Plessis [23], the knowledge management process creates new knowledge and ensures that knowledge is efficiently used in the organization. The knowledge management process facilitates other critical organizational processes, increasing the amount of knowledge required by organizational members and facilitating the rapid dissemination of organizational knowledge. Therefore, knowledge management has a profound influence on transforming the power of knowledge into innovative processes [23–25].

Regarding knowledge management in an organization, Wigg [26] introduced six phases of a knowledge management cycle: (1) creating; (2) providing; (3) synthesizing; (4) transforming; (5) disseminating; and (6) applying knowledge. Martín-de Castro et al. [27] conducted a quantitative survey with a sample of 221 firms in Finland, Russia, and China, noting that the knowledge management process includes knowledge creation, knowledge storage, knowledge sharing, and knowledge acquisition. According to their research, these factors have positive effects on innovation. In addition, they found that the knowledge creation process plays a role in the relationship between the three remaining information management processes and innovation. Pinho et al. [28] identified both barriers and opportunities through a lens of technology, society, organizations, and individuals. Their research proposed four processes of knowledge management: (1) acquiring; (2) creating; (3) sharing; and (4) transferring knowledge. However, Wee and Chua [29] adjusted the knowledge management model to propose a three-factor model with the following: (1) creating; (2) sharing; and (3) reusing knowledge.

Bigliardi et al. [30] stated that there is no single common knowledge management system for all organizations, which means organizations will adopt different knowledge management processes. Their work [30] discovered two similarities between 14 surveyed participants. First, the knowledge management process was cyclical in all surveyed organizations. Second, the surveyed organizations applied the following knowledge management processes: (1) creating, seeking, and capturing knowledge; (2) organizing, storing, and conserving knowledge; (3) distributing, transferring, and sharing knowledge; and (4) offering feedback.

García-Fernández [31] analyzed knowledge management processes based on the analysis of 78 research papers, finding processes and factors explored by researchers. First, their work identified knowledge creation as information acquisition, information dissemination, and knowledge sharing. Second, they focused on knowledge transfer and storage within the organization, including the application and use of knowledge in group work and commitment to knowledge. Meanwhile, Kianto et al. [32] analyzed the knowledge management model with the following five steps: (1) acquiring; (2) sharing; (3) creating; (4) encrypting; and (5) retaining knowledge.

Knowledge creation is an organization's ability to generate useful ideas and solutions. This step relates to various aspects of an organization's operations, including products, technological processes, and management practice. Encrypting knowledge includes activities to convert unexplained knowledge into expressive knowledge, preserving formalized knowledge, and providing the latest knowledge registered to an organization's staff. The effectiveness of this process depends on the capacity and motivation of the workers, as well as the information and communication technology infrastructure. Retaining knowledge relates to human resource management's ability to minimize the loss of expertise within an organization [32].

Obeidat et al. [33] examined the effects of knowledge management processes (acquiring, sharing, and using knowledge) and knowledge management methods (i.e., social networks, coding, and personalization) for innovation in Jordanian consulting firms. The analysis shows a significant and positive impact of knowledge management processes on innovation in Jordanian consulting firms. In addition, the research found significant and positive impacts of methodology and personalization on innovation. The social networks approach had a significant and negative impact on innovation. Yusr et al. [34] found three factors of the knowledge-based management model that affect innovation: (1) knowledge acquisition; (2) knowledge dissemination; and (3) knowledge application.

### 2.2. Research on Knowledge Management in Universities

Regarding the knowledge management model in universities, Huang et al. [35] (1998) proposed four processes to form a culture of knowledge sharing and cooperation: (1) making knowledge tangible; (2) enhancing knowledge intensity; (3) building knowledge infrastructure; and (4) creating a driving force for the development of knowledge culture. From an academic perspective, the learning community should start at the individual level by creating partial knowledge, knowledge between departments with similar academic interests or disciplines, and networks of knowledge within institutions and other organizations or corporations.

Al-Bastaki and Shajera [36] examined the readiness of knowledge management of the University of Bahrain. In addition, Ahmadi and Ahmadi [37] analyzed the model of knowledge management at the University of Shushtar, Iran. Their studies show several knowledge management model approaches to improve output and research processes. Studies show that factors in knowledge management processes in popular universities include knowledge creation, knowledge gathering, knowledge application, and knowledge dissemination [38,39].

Tsui et al. [40] used an analysis of the knowledge management model at the Bangkok University. Islam et al. [41] studied knowledge sharing behaviors of lecturers in faculties of two public universities in Bangladesh. Their work shows that knowledge sharing, in addition to other basic factors (i.e., creating, collecting, storing, and using knowledge), significantly influences the effectiveness of a university's scientific research.

Rivera and Rivera [7], in their study of the knowledge management model in the context of higher education in Mexico, proposed a model consisting of six factors: (1) leadership; (2) culture; (3) structure; (4) human resources; (5) information technology; and (6) measurement. These factors facilitate in the creation, storage, transfer, and application of studies. The authors designed a 53-question survey to apply to 36 individuals who manage knowledge through the development and implementation phases. The study showed that cultural, human, and structural aspects play important roles in university knowledge management models.

Naser et al. [5] studied the development of the knowledge management model at Al-Azhar universities and Al-Quds open universities in the Gaza Strip, Palestine. Their overall findings show that the process, leadership, people, and results of knowledge management affect the efficiency of scientific research. However, Masa'deh et al. [42], in a study on the impact of knowledge management on job performance in higher education institutions, showed that knowledge management does not impact or has a negative impact on employee satisfaction, which leads to reduced productivity at work. These results illustrate a controversy among scientists according to a common logic that when knowledge management is applied, universities will increase productivity regarding scientific research. However, the research results have shown a negative impact.

In recent years, lectures have become key in the national development strategy's educational goals [43,44]. Do et al. [45] affirmed that universities evaluate lecture performance based on their important role in research and teaching activities. In addition, a technical support process was developed to evaluate lecture performance, applying a new dynamic fuzzy technique for order preference by similarity to ideal solution (DFTOPSIS). The university functions as both a training facility and a scientific research center, encouraging lecturers to undertake scientific research activities [46]. Do et al. [47] proposed an extension of the generalized fuzzy multicriteria decision-making (MCDM) approach to evaluate a lecturer's research capacity. The results enrich the knowledge of generalized fuzzy numbers in decision-making applications, showing that the AHP method can be integrated with the proposed MCDM approach to determine the importance of criteria in the future.

### 2.3. Proposed Model

Based on previous studies on the knowledge management model in organizations and universities, this study applies a SLR, following the guidelines of Tranfield et al. [48],

Kitchenham and Charters [49], and Okoli and Schabram [50]. Based on the documents, SLR involves activities such as planning (identifying research questions), implementation (document retrieval, research selection, and data aggregation), and reporting (writing a report). Research activities are conducted through a search of raw data using keywords such as knowledge management, knowledge management model, or knowledge management model in universities. The research overview included five academic database systems: (1) ResearchGate; (2) ScienceDirect; (3) Elsvesier; (4) Scopus; and (5) Emerald Insight.

Once the articles were searched in the first phase, the title-based selection process resulted in a total of 160 relevance articles. These articles were reviewed and classified as "article found." In the second phase, summaries from each relevant article were evaluated to answer the research question. Duplicate and irrelevant articles were rejected, which left 112 "relevant articles." Filtration continued in the third phase. The articles that were used and read in detail to answer the research question were classified as "selected articles." Finally, 30 articles were chosen to aggregate the data after conducting the exclusion criteria and screening for detailed and full-text summaries. The study's proposed factors of the knowledge management model are included in Table 1 and Figure 1.

**Table 1.** Factors of the knowledge management model.

| No. | Factors | Code | Definition |
|---|---|---|---|
| 1 | Knowledge sharing | A1 | Knowledge sharing is the exchange of knowledge (skills, experience, and understanding) between individuals in an organization [40]. |
| 2 | Knowledge collection | A2 | Knowledge collection is the acquisition of knowledge. It also includes natural categories such as knowledge blocks, learners, and cultural transmission, which require a reconceptualization of knowledge as cultural and social products. These social and cultural activities are based on knowledge organization, teachers' knowledge, and action programs [51]. |
| 3 | Knowledge assessment | A3 | Knowledge assessment is an indispensable factor of the knowledge management model. This factor must be located in the first stage of the knowledge management process. Knowledge assessment demonstrates and validates the qualification of the acquired knowledge. In turn, it illustrates knowledge that can be used, shared, or stored within an organization [52]. |
| 4 | Knowledge application | A4 | Knowledge application is the process of presenting and applying knowledge that is collected and authenticated to influence decision making, design policy, problem solving, or creating solutions for human needs. This factor takes advantage of opportunities to create new knowledge. Knowledge must go through the process of building, transforming, and maintaining in the process of using and acting [53]. |
| 5 | Knowledge creation | A5 | Knowledge creation is an organization's ability to generate new and useful ideas and solutions. This factor relates to various aspects of an organization's operations, including product and process technology and reality of management [54]. |
| 6 | Knowledge storage | A6 | Knowledge storage is the use of technology to provide a means of storing and retrieving knowledge through computerization [55]. This makes the knowledge accessible to others or the next generation. |

**Table 1.** *Cont.*

| No. | Factors | Code | Definition |
|---|---|---|---|
| 7 | Leadership factors | A7 | Leadership factors in knowledge management are related to the mission and vision, academic environment, empowerment, management system, openness to change, and policies that motivate scientific research [51,56]. |
| 8 | Knowledge management with big data system | A8 | Knowledge management is created through the analysis of big data systems. Its integration with solid knowledge systems ensures that data can be analyzed and classified into useful information and translated into knowledge [33,57]. |

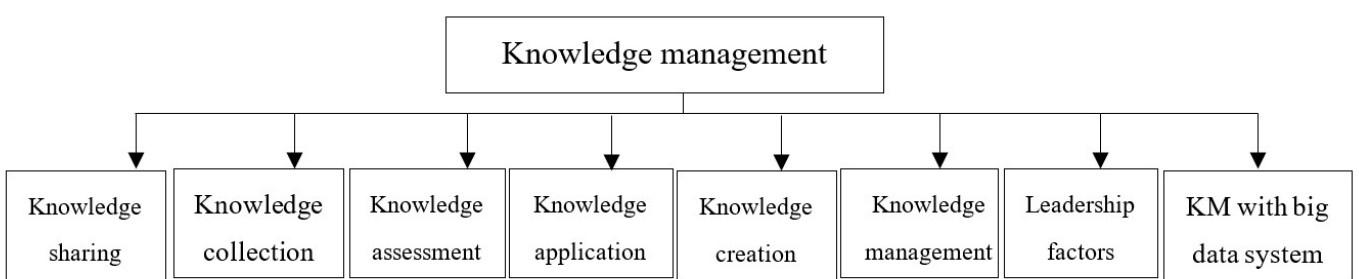

**Figure 1.** Research model.

## 3. Research Method

### 3.1. FAHP Method

The pure AHP method tends to be less effective when dealing with uncertainty in the decision-making process. This leads to the development of FAHP methods, which has been used by many researchers to solve decision-making problems in various areas. The FAHP deals with the uncertainty and imprecision of the service evaluation process [58]. Several FAHP methods are available for solving managerial problems. The method proposed by Buckley [59] and Hsieh et al. [60] is used to calculate factor weights in works by Sun [61] and Owusu-Agyeman et al. [62].

3.1.1. Step 1: Making Pair-wise Comparisons and Obtaining the Individual Judgment Matrices

The experts make pair-wise comparisons of the importance or preference between each pair of criteria. The comparison of criterion is in the form of linguistic variables. This can be achieved through questionnaires. A linguistic variable's values are words or sentences in a natural or artificial language. In this paper, triangular fuzzy numbers (TFNs) represent subjective pair-wise comparisons of decision makers, namely "equally important," "moderately more important," "strongly more important," "very strongly more important," and "absolutely more important."

$$A^k = \begin{bmatrix} a_{11}^k & a_{12}^k & \dots & a_{1n}^k \\ a_{21}^k & a_{22}^k & \dots & a_{2n}^k \\ \dots\dots\dots\dots\dots \\ a_{m1}^k & a_{m2}^k & a_{mn}^k \end{bmatrix} \tag{1}$$

The evaluation of experts is calculated according to Equation (2). The derived matrix is in the form of Equation (3).

$$\widetilde{a}_{ij} = \left( \widetilde{a}_{ij}^1 \otimes \widetilde{a}_{ij}^2 \dots . \otimes \widetilde{a}_{ij}^{10} \right) \tag{2}$$

$$
A = \begin{bmatrix}
a_{11} & a_{12} & \dots & a_{1n} \\
a_{21} & a_{22} & \dots & a_{2n} \\
\multicolumn{4}{c}{\dots\dots\dots\dots} \\
a_{m1} & a_{m2} & & a_{mn}
\end{bmatrix} \tag{3}
$$

### 3.1.2. Step 2: Constructing the Comparison Matrices

Here, the geometric mean method is used to establish the representative comparison fuzzy matrix for group decisions. Experts make a pair-wise comparison of criteria. A comparison matrix is a result of the pair-wise comparison.

Geometric mean and fuzzy weight of factors are calculated according to Hsieh et al. [53]:

$$
\widetilde{r}_i = (\widetilde{a}_{i1} \otimes \dots \otimes \widetilde{a}_{ij} \otimes \dots \otimes \widetilde{a}_{in})^{1/n} \tag{4}
$$

$$
\widetilde{w}_i = \widetilde{r}_i \otimes [\widetilde{r}_i \oplus \dots \oplus \widetilde{r}_i \oplus \dots \oplus \widetilde{r}_i]^{-1} \tag{5}
$$

where $\widetilde{a}_{ij}$ represents a relative importance of criterion i to j and $\widetilde{r}_i$ is the geometric mean value of criterion i. $\widetilde{w}_i$ is the fuzzy weight of criterion i, represented by TFN. $\widetilde{w}_i = (lw_i, mw_i, uw_i)$ $lw_i$, $mw_i$, and $uw_i$ are the low, middle, and high value of TFN.

### 3.1.3. Step 3: Defuzzification

Defuzzification is the process of converting a fuzzy number to a crisp number. The most used defuzzification method is the center of area (COA) method (or centroid method). This method determines the COA of a fuzzy set and returns the corresponding crisp value. The COA method is used to calculate the best nonfuzzy performance (*BNP*) value for each factor.

$$
BNP = \frac{[(U_{wi} - L_{wi}) + (M_{Wi} - L_{wi})]}{3} + L_{wi} \tag{6}
$$

### 3.1.4. Step 4: Calculating Consistency Ratio for Matrix

Defuzzification is applied to calculate the consistency ratio index. Next, the consistency matrix is obtained as in the pure AHP. The two matrices, $A_m$ and $A_g$, are derived from the comparison matrix using the defuzzification method by Gogus and Boucher [63].

$A_m$ is the matrix derived from the mean (m) values of fuzzy comparison matrix:

$$
A_m \, [a_{ijm}] \tag{7}
$$

$A_g$ is the matrix derived from the geometric mean by the smallest value (l) and the largest possible (m):

$$
A_g = \sqrt[2]{a_{iju}} a_{ijl} \tag{8}
$$

The two matrices with the value of consistency ratio (CR) below 0.1 indicate the consistency of the matrix.

The defuzzification is achieved using the following equation:

$$
A = [a_{ijl} + 2 \times a_{ijm} + a_{iju}/4] \tag{9}
$$

Then, the matrix is normalized with the following equation:

$$
A_i = \left[ \frac{a_i}{\sum_{i=1}^{n} a_i} \right] \tag{10}
$$

The consistency index (CI), for a comparison matrix, can be computed with the use of the following equation:

$$
CI = (\lambda_{max} - n)/(n - 1) \tag{11}
$$

where $\lambda_{max}$ is the largest eigenvalue of the comparison matrix and $n$ is the dimension of the matrix.

$$\lambda_{max} = \frac{1}{n} \sum_{i=1}^{n} (AW)_i / w_i \tag{12}$$

The consistency ratio CR is defined as a ratio between the consistency of a given evaluation matrix and consistency of a random matrix:

$$CR = (CI/RI) \tag{13}$$

where $RI(n)$ is a random index (RI) that depends on $n$, as shown in Table 2.

**Table 2.** RI. of random matrices.

| n | 1 | 2 | 3 | 4 | 5 | 6 | 7 | 8 | 9 | 10 |
|---|---|---|---|---|---|---|---|---|---|----|
| **R** | 0 | 0 | 0.52 | 0.90 | 1.12 | 1.24 | 1.32 | 1.41 | 1.45 | 1.49 |

If the C.R. of a comparison matrix is equal to or less than 0.1, it may be acceptable. If the C.R. is unacceptable, the decision maker is encouraged to repeat the pair-wise comparisons.

*3.2. Linguistic Variables and Fuzzy Scales*

The decision makers make pair-wise comparisons of the importance of (or their preference for) each pair of factors. The comparison of one factor over another is in the form of linguistic variables, which can be done with questionnaires. A linguistic variable's values are words or sentences in a natural or artificial language.

In this paper, TFNs represent subjective pair-wise comparisons of decision makers. The linguistic variables and fuzzy scales for importance are used to convert linguistic variables into TFNs (see Table 3).

**Table 3.** Linguistic variables and their fuzzy number.

| Numerical Rating | Linguistic Variable | TFN |
|:---:|:---:|:---:|
| 1 | Equally important | (1,1,1) |
| 2 | Intermediate value between 1 and 3 | (1,2,3) |
| 3 | Moderately important | (2,3,4) |
| 4 | Intermediate value between 3 and 5 | (3,4,5) |
| 5 | Strongly important | (4,5,6) |
| 6 | Intermediate value between 5 and 7 | (5,6,7) |
| 7 | Very strongly important | (6,7,8) |
| 8 | Intermediate value between 7 and 9 | (7,8,9) |
| 9 | Extremely important | (8,9,10) |

Twelve experts, including managers and scholars from seven universities, participated in the survey. Four of the returned questionnaires were invalid; they were returned to the experts for revisions. Two responses were excluded from the analysis because the experts refused to correct their answers. The results of this study are based on 10 experts. Table 4 presents the experts' information.

**Table 4.** Basic information of respondents.

| No. | Position | Years of Experience | University Type |
|---|---|---|---|
| 1 | Vice-rector | 10 | Public |
| 2 | Rector | 15 | Public |
| 3 | Editor in chief | 12 | Public |
| 4 | Vice-rector | 15 | Private |
| 5 | President | 20 | Public |
| 6 | President | 20 | Private |
| 7 | Rector | 15 | Public |
| 8 | Head of Department of Science, Technology, and International Cooperation | 10 | Public |
| 9 | Vice-rector | 20 | Public |
| 10 | Dean of Faculty | 20 | Private |

## 4. Data Analysis

Big data systems are related to the use of information technology and the advancement of the 4.0 industrial revolution. However, based on the obtained results, few studies could be found, particularly on the driving force of scientific research. The topics in the model of knowledge intelligence in universities (i.e., knowledge collection, knowledge assessment, knowledge use, knowledge sharing, and knowledge storage) are the most popular issues in the model of knowledge intelligence in universities. Therefore, this study only includes eight in the survey. The eight factors in the model of knowledge management are shown in Table 5.

**Table 5.** Factor and its code.

| No. | Code | Factor |
|---|---|---|
| 1 | A1 | Knowledge sharing |
| 2 | A2 | Knowledge collection |
| 3 | A3 | Knowledge assessment |
| 4 | A4 | Knowledge application |
| 5 | A5 | Knowledge creation |
| 6 | A6 | Knowledge storage |
| 7 | A7 | Leadership factors |
| 8 | A8 | Knowledge management with big data systems |

The importance comparison for each factor was performed via questionnaire. The importance rank of the factors was assessed by 10 experts and converted from linguistic variables to the equivalent fuzzy numbers.

The geometric mean method by Buckley [46] was used to calculate each factor in the comparison matrix.

$$\widetilde{a}_{ij} = \left( \widetilde{a}_{ij}^{1} \otimes \widetilde{a}_{ij}^{2} \ldots \otimes \widetilde{a}_{ij}^{10} \right) \tag{14}$$

Take $\widetilde{a}_{12}$ as an example:

$$\widetilde{a}_{12} = (1,1,1) \otimes (1,2,3) \otimes \ldots \otimes (1,2,3)^{1/10} = \left( (1 \times 1 \times \ldots \times 1)^{1/10}, (1 \times 2 \times \ldots \times 2)^{1/10}, (1 \times 3 \times \ldots \times 3)^{1/10} \right) = (0.25, 1.20, 2.53) \tag{15}$$

Similarly, the comparison matrix is shown in Table 6.

**Table 6.** Comparison matrix of the factors in the knowledge management model in universities.

|  | A1 | A2 | A3 | A4 | A5 | A6 | A7 | A8 |
|---|---|---|---|---|---|---|---|---|
| A1 | *1.00* 1.00 1.00 | *0.25* 1.20 2.53 | *0.25* 3.73 4.24 | *1.22* 2.88 3.14 | *0.14* 0.64 1.02 | *1.02* 2.02 3.31 | *0.20* 2.60 3.26 | *0.13* 1.43 2.01 |
| A2 | *0.25* 0.84 1.65 | *1.00* 1.00 1.00 | *0.20* 2.16 3.20 | *1.12* 2.13 3.21 | *0.17* 0.87 1.25 | *0.17* 2.20 3.02 | *0.13* 0.65 1.04 | *0.11* 0.33 1.45 |
| A3 | *0.13* 0.27 1.20 | *0.17* 0.46 1.84 | *1.00* 1.00 1.00 | *0.17* 0.77 1.52 | *0.13* 1.43 2.10 | *0.25* 2.47 3.21 | *0.17* 0.62 1.35 | *0.13* 0.22 1.53 |
| A4 | *0.17* 0.35 1.50 | *0.13* 0.32 1.29 | *0.20* 1.30 3.88 | *1.00* 1.00 1.00 | *0.20* 1.02 2.55 | *0.33* 2.62 3.16 | *0.25* 1.45 2.42 | *0.17* 0.44 1.68 |
| A5 | *0.20* 1.57 3.21 | *0.17* 1.15 2.15 | *0.11* 0.70 1.55 | *0.13* 0.53 1.47 | *1.00* 1.00 1.00 | *1.00* 3.97 4.09 | *0.33* 2.50 3.18 | *0.20* 0.93 2.50 |
| A6 | *0.11* 0.25 1.62 | *0.14* 0.45 1.20 | *0.13* 0.41 1.84 | *0.13* 0.38 1.43 | *0.11* 0.25 1.03 | *1.00* 1.00 1.00 | *0.13* 0.90 1.02 | *0.13* 0.54 1.52 |
| A7 | *0.13* 0.39 1.36 | *0.17* 1.55 2.33 | *0.20* 1.60 2.99 | *0.17* 0.69 1.26 | *0.13* 0.40 1.21 | *0.13* 1.11 2.27 | *1.00* 1.00 1.00 | *0.17* 0.58 1.36 |
| A8 | *0.17* 0.70 1.72 | *1.33* 3.05 4.11 | *2.20* 3.55 5.23 | *0.20* 1.27 2.23 | *0.17* 1.08 1.88 | *1.25* 1.26 2.81 | *0.17* 1.02 1.49 | *1.00* 1.00 1.00 |

Next, the fuzzy weight for each factor was calculated. The following formula was used for the calculation ($\tilde{r}_1$):

$$\tilde{r}_1 = (\tilde{a}_{11} \otimes \tilde{a}_{12} \otimes \tilde{a}_{13} \otimes \tilde{a}_{14} \otimes \tilde{a}_{15} \otimes \tilde{a}_{16} \otimes \tilde{a}_{17} \otimes \tilde{a}_{18})^{1/8} = \left( (1 \times 0.25 \times \ldots \times 0.13)^{1/8}, (1 \times 1.20 \times \ldots \times 1.43)^{1/8}, \right.$$
$$\left. (1 \times 2.53 \times \ldots \times 2.01)^{1/8} \right) = (0.36, 1.67, 2.29) \tag{16}$$

Similarly, the $\tilde{r}_i$ was obtained as follows:

$$\tilde{r}_2 = (0.26, 1.06, 1.77); \ \tilde{r}_3 = (0.20, 0.68, 1.62); \ \tilde{r}_4 = (0.24, 0.84, 1.99); \ \tilde{r}_5 = (0.27, 1.26, 2.18); \ \tilde{r}_6 = (0.16, 0.46, 1.30);$$
$$\tilde{r}_7 = (0.19, 0.80, 1.61); \ \tilde{r}_8 = (0.49, 1.38, 2.25) \tag{17}$$

Then, the factor weight ($\tilde{w}_i$) for each factor was calculated with the following equation:

$$\tilde{w}_1 = \tilde{r}_1 \otimes [\tilde{r}_3 \oplus \tilde{r}_2 \oplus \tilde{r}_4 \oplus \tilde{r}_5 \oplus \tilde{r}_6 \oplus \tilde{r}_7 \oplus \tilde{r}_8)^{-1} = ((0.26, 1.06, 1.77) \otimes (1/1.77 + \ldots + 2.25),$$
$$1/(1.06 + \ldots + 1.38), \ 1/(0.26 + \ldots + 0.49)) = (0.02, 0.21, 1.05) \tag{18}$$

Similarly, $\tilde{w}_i$ for each factor was as follows:

$$\tilde{w}_2 = (0.02, 0.13, 0.81); \ \tilde{w}_3 = (0.01, 0.08, 0.74); \ \tilde{w}_4 = (0.02, 0.10, 0.91); \ \tilde{w}_5 = (0.02, 0.15, 1.00); \ \tilde{w}_6 = (0.01, 0.06, 0.60);$$
$$\tilde{w}_7 = (0.01, 0.10, 0.74); \ \tilde{w}_8 = (0.03, 0.17, 1.03); \tag{19}$$

Using the COA method, *BNP* values in fuzzy number form for each factor are obtained as follows. Take *BNP* value for A1 factor (knowledge sharing) as an example, using the following equation:

$$BNP = \frac{[(U_{w1} - L_{w1}) + (M_{W1} - L_{w1})]}{3} + L_{w1} = \frac{[(1.05 - 0.02) + (0.21 - 0.02)]}{3} + 0.02 = 0.427 \tag{20}$$

After having the crisp number of the *BNP* value, the normalization was achieved using Equation (9). The obtained matrix is shown in Table 7. Using Equation (10), the normalized matrix is derived as in Table 8. The other *BNP* values for each factor are shown in Table 9.

**Table 7.** Defuzzification matrix.

|  | A1 | A2 | A3 | A4 | A5 | A6 | A7 | A8 |
|---|---|---|---|---|---|---|---|---|
| A1 | 1.00 | 1.29 | 2.99 | 2.53 | 0.61 | 2.09 | 2.16 | 1.25 |
| A2 | 0.89 | 1.00 | 1.93 | 2.15 | 0.79 | 1.90 | 0.61 | 0.55 |
| A3 | 0.47 | 0.73 | 1.00 | 0.81 | 1.27 | 2.10 | 0.69 | 0.52 |
| A4 | 0.59 | 0.51 | 1.67 | 1.00 | 1.20 | 2.19 | 1.39 | 0.68 |
| A5 | 1.64 | 1.15 | 0.76 | 0.66 | 1.00 | 3.26 | 2.13 | 1.14 |
| A6 | 0.56 | 0.56 | 0.69 | 0.58 | 0.41 | 1.00 | 0.74 | 0.68 |
| A7 | 0.56 | 1.40 | 1.60 | 0.70 | 0.53 | 1.16 | 1.00 | 0.67 |
| A8 | 0.82 | 2.89 | 3.63 | 1.24 | 1.05 | 1.64 | 0.92 | 1.00 |

**Table 8.** Normalize matrix.

|      | A1   | A2   | A3   | A4   | A5   | A6   | A7   | A8   | Geometric Mean |
|------|------|------|------|------|------|------|------|------|------|
| **A1** | 0.15 | 0.14 | 0.21 | 0.26 | 0.09 | 0.14 | 0.22 | 0.19 | 0.18 |
| **A2** | 0.14 | 0.10 | 0.14 | 0.22 | 0.12 | 0.12 | 0.06 | 0.09 | 0.12 |
| **A3** | 0.07 | 0.08 | 0.07 | 0.08 | 0.19 | 0.14 | 0.07 | 0.08 | 0.10 |
| **A4** | 0.09 | 0.05 | 0.12 | 0.10 | 0.17 | 0.14 | 0.14 | 0.10 | 0.12 |
| **A5** | 0.25 | 0.12 | 0.05 | 0.07 | 0.15 | 0.21 | 0.22 | 0.18 | 0.16 |
| **A6** | 0.09 | 0.06 | 0.05 | 0.06 | 0.06 | 0.07 | 0.08 | 0.10 | 0.07 |
| **A7** | 0.09 | 0.15 | 0.11 | 0.07 | 0.08 | 0.08 | 0.10 | 0.10 | 0.10 |
| **A8** | 0.13 | 0.30 | 0.25 | 0.13 | 0.15 | 0.11 | 0.10 | 0.15 | 0.17 |

**Table 9.** Rankings and local weights of factor.

| Factor | Weight ($\tilde{w}$) | *BNP* Value | Geometric Mean (GM) | Rank |
|--------|--------|--------|--------|------|
| Knowledge sharing | (0.02, 0.21, 1.05) | 0.427 | 0.175 | 1 |
| Knowledge collection | (0.02, 0.21, 1.05) | 0.320 | 0.123 | 5 |
| Knowledge assessment | (0.01, 0.08, 0.74) | 0.280 | 0.097 | 7 |
| Knowledge application | (0.02, 0.10, 0.91) | 0.345 | 0.116 | 4 |
| Knowledge creation | (0.02, 0.15, 1.00) | 0.391 | 0.156 | 3 |
| Knowledge storage | (0.01, 0.06, 0.60) | 0.222 | 0.070 | 8 |
| Leadership factors | (0.01, 0.10, 0.74) | 0.283 | 0.097 | 6 |
| Knowledge management with big data system | (0.03, 0.17, 1.03) | 0.412 | 0.165 | 2 |

The consistency indices were calculated using Equations (11)–(13). The obtained results are:

$$\lambda = 8.970; \text{CI.} = 0.138, \text{ with } n = 8; \text{RI} = 1.41 \text{ CR} = \text{CI/RI} = 0.0983 \qquad (21)$$

After obtaining the factor weight ($\tilde{w}$), *BNP* value, and geometric mean matrix for each factor, the order rank of factors is shown in Table 9.

Table 9 shows *BNP* importance weight and the geometric mean of eight factors in the model of knowledge management at universities in Hanoi, Vietnam. The results in Table 9 reveal the experts' experience and knowledge of the priority order of the selected factors in the knowledge management research model. Knowledge sharing ranks first with the highest *BNP* value (*BNP* = 0.427; GM = 0.175). Knowledge management places second with big data systems (*BNP* = 0.412, GM = 0.165); knowledge creation is third (*BNP* = 0.391; GM = 0.165). The fourth ranked factor is knowledge application (*BNP* = 0.345; GM = 0.116), followed by knowledge collection in fifth (*BNP* = 0.280; GM = 0.097). Leadership ranks sixth (*BNP* = 0.283; GM = 0.097), knowledge assessment ranks seventh (*BNP* = 0.280; GM = 0.097), and, finally, the eighth place belongs to knowledge storage (*BNP* = 0.0.222; GM = 0.070).

Based on the obtained result, three factors (i.e., knowledge sharing, knowledge management with big data systems, and knowledge creation) are the most important in the knowledge management model. The second group of influential factors includes knowledge application and knowledge gathering. Finally, the less important factors are leadership, assessing knowledge, and storing knowledge.

This finding shows an interesting implication that the most influential factors are knowledge sharing, creation, and application of knowledge management with the use of

science and technology derived from the industrial revolution 4.0. However, according to recent studies, these factors only emerged in the field of information management in the past five years [5,59,60,64]. In addition, knowledge creation is assessed as high priority, demonstrating the leading role of universities in knowledge creation, science, and technology innovation. In the new context, the other descending factors—the use and collection of knowledge—are important factors because a university's mission is to gather and transmit knowledge to learners. The less influential factors are leadership, storage, and evaluation of knowledge. This finding also shows the specific goals of most universities in Vietnam. It is believed that universities should prioritize the core issues of knowledge governance, such as creation, application, big data issues, and knowledge sharing. This is reasonable and suitable to the limited resources of universities as Vietnam promotes innovative activities and takes advantage of the achievements of the 4.0 industrial revolution to develop science and technology.

## 5. Conclusions

The objective of this study is to explore and evaluate the factors of the knowledge management model at universities in Hanoi, Vietnam. In this study, the authors presented an overview of the current knowledge management model and the knowledge management model in universities. First, the study conducted a SLR of papers published in the last decade. A total of 160 related papers were found; 112 papers were used in the research. Finally, 30 papers were used to identify the correlation between factors of the knowledge management model. Second, factors were evaluated and ranked using the FAHP method. This included the opinions of 10 experts from different universities.

The obtained results by the SLR method show the first factors proposed for the knowledge management model in universities through the literature review. By constructing a correlation matrix, two groups of factors were identified with appearance order from high to low. The findings by the FAHP method show eight factors were analyzed and categorized into three groups. The first group included knowledge sharing, knowledge management with big data system, and knowledge creation. This finding is in accordance with previous studies regarding the evaluation of the model of knowledge control in universities [7,42,65]. The first group of factors applies scientific achievements of the 4.0 industrial revolution of the knowledge management model. The second group, which relates to the use and collection of knowledge, can be considered a traditional group that performs basic functions of the university. The third, less important, group includes leadership factors, knowledge assessment, and knowledge storage. This finding does not align with previous studies when main functions of universities seek activities for creating motivation to conduct research, knowledge assessment, and knowledge storage [35,54,66,67]. Based on the results, big data systems and the knowledge management method are related to the use of information technology and the achievements of the 4.0 industrial revolution [66,68]. Moreover, these findings show significant implications in knowledge management in Vietnamese universities when leadership and motivation for conducting scientific research are not at a high priority level. This could also be an important finding for future studies when looking at issues related to knowledge management at universities in Vietnam.

**Author Contributions:** N.T.P. and A.D.D. have mainly contributed for giving the paper idea and reviewing of all literatures, section and checking the typos for this manuscript. Q.V.N. and V.L.T. contributed for interviewing and collecting the data from experts, model and matrix building, questionnaire designing and result calculating. T.T.B.D., D.L.H., and X.T.H. contributed for reading and giving the comment and conclusion parts of the paper. All authors have read and agreed to the published version of the manuscript.

**Funding:** This research is funded by Vietnam National Foundation for Science and Technology Development (NAFOSTED) under grant number 503.01-2020.02.

**Conflicts of Interest:** The authors declare that they do not have any conflict of interests.

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
