# Peer review of "Research on Knowledge Management Models at Universities Using Fuzzy Analytic Hierarchy Process (FAHP)"

_sustainability, doi:10.3390/su13020809_

Round 1

Reviewer 1 Report

In this paper, the authors explore and evaluate the factors of the knowledge management model at universities in Hanoi, Vietnam. Moreover, the authors presented an overview of the current knowledge management model and the knowledge management model in universities. First, the study conducted an SLR of papers published in the last decade. A total of 160 related papers were found; 112 papers were used in the research. Finally, 30 papers were utilized to identify the correlation between factors of the knowledge management model. Second, factors were evaluated and ranked using the FAHP method. This included the opinions of 10 experts from different universities. The study is well presented and proved with mathematical FAHP steps. However, the following concerns need to be addressed before final publication.

  1. Please mention the corresponding author in the paper and his affiliation. You did not mention any of the authors as the corresponding. At least there should be one corresponding author.
  2. It will have a good impression if the authors provide some details of the results in the abstract i.e., the results achieved.
  3. In good papers, most recent literature provides an insight to the readers about the current trends of the methods and techniques. Please provide/cite some of the recent literature on the applications of fuzzy decision making in various fields e.g., it is used in SDN for controller selection and controller placement.
  • https://journals.plos.org/plosone/article?id=10.1371/journal.pone.0217631 (controller selection for SDN).
  • https://dl.acm.org/doi/abs/10.1145/3307334.3328617 (Controller place selection in SDN)

Similarly, mention at least 5 recent papers (Published in the last 3 years) including the above on the utilization of fuzzy decision-making methods.

  1. Provide a figure workflow diagram for the proposed model in section 2.3
  2. Please show the consistency index table from the comparison matrix as in line#230 you have stated “The two matrices with the value of CR below 0.1 indicate the consistency of the matrix.”. Hence, show the consistency index of the comparison matrices in a table.
  3. Provide the author's contributions at the end of the manuscript. i.e., the contribution of each author in the paper.

Author Response

Point 1: Please mention the corresponding author in the paper and his affiliation. You did not mention any of the authors as the corresponding. At least there should be one corresponding author.

Response 1: Thank you very much for your suggestion. The corresponding author in the paper and his affiliation has been added as Reviewer’s suggestion.

Point 2: It will have a good impression if the authors provide some details of the results in the abstract i.e., the results achieved.

Response 2: Thank you very much for your comments.The results ranking of factors have been available in abstract.

Point 3: Please provide/cite some of the recent literature on the applications of fuzzy decision making in various fields e.g., it is used in SDN for controller selection and controller placement.

Response 3: Thank you very much for your suggestion. Some of the recent literature on the applications of fuzzy decision making in various fields have been added as Reviewer’s suggestion such as:  Jehad Ali et. al (2019) in SDN for controller selection and controller placement ; Lyu et. al., ( 2020) in risk assessment; Stanković et. Al., (2019) in traffic accessibility; Sani et.al., (2019) in Knowledge management; Beyca et.al (2020) in determine the tacit knowledge criteria.

Point 4: Provide a figure workflow diagram for the proposed model in section 2.3

Response 4: Thank you very much for your comments. Ffigure workflow diagram for the proposed model have been added in section 2.3 as Reviewer’s suggestion

Point 5: Please show the consistency index table from the comparison matrix

Response 5:  Thank you very much for your comments. As formular (21) in line 309 have showed that

            λ = 8.970; CI. = 0.138 => with n = 8 => RI = 1.41    CR = CI/RI = 0.0983

So it can conclude that The two matrices (Comparison matrix and Defuzzification matrix) with the value of CR below 0.1 indicate the consistency of the matrix

Point 6: Provide the author's contributions at the end of the manuscript. i.e., the contribution of each author in the paper

Response 6:  Thank you very much for your comments. Author's contributions have been added at the end of the manuscript as Reviewer’s suggestion

The authors would like to thank again the reviewers for the time and expertise they have invested in these reviews. The revised manuscript with marked changes has been resubmitted to your journal. We look forward to your positive response.

Sincerely,

Anh Duc DO

Reviewer 2 Report

Dear Authors

The article is constructed at a high scientific level. Apart from a few minor shortcomings in my subjective opinion, the article is very good. Very methodologically well prepared. Current professional well-matched literature. I recommend publishing an article with a request for verification. Below are some comments asking for their consideration. Formula (5) does not look very good All tables are too wide. Shouldn't the line before 288 say: SUM for A8? Not specified in the table for 6.53, 9.54, 14.27, 9.67, 6.87, 15.33, 9.65, 6.50 I suggest that these references be removed from the summary.

I wish you good health

Greetings

Reviewer

Author Response

Point 1: Formula (5) does not look very good

Response 1: Thank you very much for your comments. Formula (5) are revised as standard as Reviewer’s suggestion

Point 2: All tables are too wide

Response 2: Thank you very much for your comments. As the format of publisher so if its required to narrow it the authors will revised as Reviewer’s suggestion

 Point 3: Shouldn't the line before 288 say: SUM for A8? Not specified in the table for 6.53, 9.54, 14.27, 9.67, 6.87, 15.33, 9.65, 6.50 I suggest that these references be removed from the summary

Response 3: Thank you very much for your comments. These references have be removed from the summary as Reviewer’s suggestion

The authors would like to thank again the reviewers for the time and expertise they have invested in these reviews. The revised manuscript with marked changes has been resubmitted to your journal. We look forward to your positive response.

Sincerely,

Anh Duc DO